# Ca^2+^-Dependent and -Independent Calmodulin Binding to the Cytoplasmic Loop of Gap Junction Connexins

**DOI:** 10.3390/ijms24044153

**Published:** 2023-02-19

**Authors:** Oanh Tran, Silke Kerruth, Catherine Coates, Hansween Kaur, Camillo Peracchia, Tom Carter, Katalin Török

**Affiliations:** 1Molecular and Clinical Sciences Research Institute, St. George’s University of London, Cranmer Terrace, London SW17 0RE, UK; 2Department of Pharmacology and Physiology, University of Rochester Medical Center, 601 Elmwood Avenue, Rochester, NY 14642, USA

**Keywords:** connexin, calmodulin, calcium, peptide, fluorescence, kinetics, gap junction

## Abstract

Ca^2+^/calmodulin (Ca^2+^/CaM) interaction with connexins (Cx) is well-established; however, the mechanistic basis of regulation of gap junction function by Ca^2+^/CaM is not fully understood. Ca^2+^/CaM is predicted to bind to a domain in the C-terminal portion of the intracellular loop (CL2) in the vast majority of Cx isoforms and for a number of Cx-s this prediction has proved correct. In this study, we investigate and characterise both Ca^2+^/CaM and apo-CaM binding to selected representatives of each of the α, β and γ connexin family to develop a better mechanistic understanding of CaM effects on gap junction function. The affinity and kinetics Ca^2+^/CaM and apo-CaM interactions of CL2 peptides of β-Cx32, γ-Cx35, α-Cx43, α-Cx45 and α-Cx57 were investigated. All five Cx CL2 peptides were found to have high affinity for Ca^2+^/CaM with dissociation constants (*K*_d(+Ca)_) from 20 to 150 nM. The limiting rate of binding and the rates of dissociation covered a broad range. In addition, we obtained evidence for high affinity Ca^2+^-independent interaction of all five peptides with CaM, consistent with CaM remaining anchored to gap junctions in resting cells. However, for the α-Cx45 and α-Cx57 CL2 peptides, Ca^2+^-dependent association at resting [Ca^2+^] of 50–100 nM is indicated in these complexes as one of the CaM Ca^2+^ binding sites displays high affinity with *K*_d_ of 70 and 30 nM for Ca^2+^, respectively. Furthermore, complex conformational changes were observed in peptide-apo-CaM complexes with the structure of CaM compacted or stretched by the peptide in a concentration dependent manner suggesting that the CL2 domain may undergo helix-to-coil transition and/or forms bundles, which may be relevant in the hexameric gap junction. We demonstrate inhibition of gap junction permeability by Ca^2+^/CaM in a dose dependent manner, further cementing Ca^2+^/CaM as a regulator of gap junction function. The motion of a stretched CaM–CL2 complex compacting upon Ca^2+^ binding may bring about the Ca^2+^/CaM block of the gap junction pore by a push and pull action on the CL2 C-terminal hydrophobic residues of transmembrane domain 3 (TM3) in and out of the membrane.

## 1. Introduction

Connexins (Cx) are membrane-spanning proteins that form gap junctions, which allow the exchange of small molecules between cells, e.g., ATP, IP_3_ and ions such as Ca^2+^. Gap junction permeability is a critical factor for the precision of intercellular communications, such as the wave-like flux of second messengers and synchronicity of cell response in cardiac muscle. Voltage and pH are well-characterised regulators of gap junction permeability. In addition, gap junction function is modulated by interaction with a broad range of proteins [1]. Binding site motifs for the ubiquitous Ca^2+^ regulatory protein calmodulin (CaM) have been identified in a number of Cx-s. Thus, Ca^2+^/CaM was put forward as a modulator of gap junction function ([2,3], reviewed in [4,5] (Figure 1)). A CaM binding domain was first described in the β-Cx32 C-terminal tail (CT), on the basis of its propensity for folding into an amphipathic α-helix [2,3]. In addition to the CT peptide, which bound with a low μM *K*_d_, a high affinity CaM binding site (nM *K*_d_) was identified in the N-terminal (NT) domain of β-Cx32 [6,7]. More CaM binding sites were identified in CT-s of γ-Cx34.7, γ-Cx35 and γ-Cx36 [8]. Although a role for the second half of the intracellular loop of Cx-s in gap junction gating had been put forward earlier [9,10], identification of the second half of the cytoplasmic loop (CL2) as a third Ca^2+^/CaM binding domain and demonstration of concomitant inhibition by Ca^2+^/CaM binding of gap junction coupling for α-connexins Cx43, Cx44, Cx45 and Cx50 represented a breakthrough [11,12,13,14,15]. CaM binding to one or more domains of almost all Cx isoforms is strongly predicted by a sequence scoring algorithm [16] (Table 1). In addition to CaM binding to the NT of β-Cx 26, the CT of β-Cx30, γ-Cx40 and α-Cx57, binding is predicted to the CL2 of type β connexins Cx26r, Cx31r, Cx31.1r, Cx32r; of type γ connexins Cx33r, Cx34.7, Cx35, Cx36r, Cx37r, Cx40r (though not Cx40d, c, h or m); and of type α connexins Cx43r, Cx44, Cx45m, Cx46r, Cx50m, Cx56 and Cx57r. Further references to the localisation of putative and established CaM binding sites of Cx isoforms are provided in Table 1.

As there is good sequence similarity of the predicted CL2 CaM binding regions within each family of isoforms—but marked differences between families—we selected representative Cx-s from each family for analysis. We used fluorescence spectroscopy and stopped-flow fluorimetry at physiological ionic strength, and pH 7.5 and 20 °C, to study the interactions of synthetic CL2 peptides derived from β-Cx32, γ-Cx35, α-Cx43, α-Cx45 and α-Cx57 with Ca^2+^/CaM and apo-CaM. β-Cx32 is an important hepatic isoform [17]; γ-Cx35 is present in vertebrate retina [18]; β-Cx43 and α-Cx45 are important in aortic smooth muscle and cardiac conduction system [19,20]; and α-Cx57 is also present in the retina [21]. To study the well-established Ca^2+^.CaM block of gap junctions [22], we used human umbilical cord endothelial cells (HUVEC) in which Cx43 is predominantly expressed, but a number of other isoforms are also present [23], to demonstrate inhibition of gap junction permeability by Ca^2+^.CaM and the abolition of the inhibition with the cell-permeant high affinity Ca^2+^/CaM binding mTrp peptide [24]. With these approaches, we aimed to gain further insights into the mechanism of the interaction between CaM and Cx-s, in different conditions, in resting cells and stimulated cells.

**Table 1 ijms-24-04153-t001:** Predicted and/or identified CaM binding domains of Cx isoforms.

Type	Cx	NH_2_-Terminal Tail (NT)	Cytoplasmic Loop’s Second Half (CL2)	C-Terminal Tail (CT)	Localisation
**β**	26	VN**KHSTSIGKIWLTVLFIFR**IM	IKTQ**KVRIEGSLWWTYTTSIFFR**V		cochlea [25], epithelium [26]
**β**	30		IKRQ**KVRIEGSLWWTYTSSI**FFRI	VAEL**CYLLLKLCFRRSKRT**QAQR	brain [27]
**β**	31		YSHPG**KKHGGLWWTYLFSLIFK**LII		placenta, skin, eye [28]
**β**	31.1		YPNPGK**KRGGLWWTYVCSLL**FKATID		skin [29]
**β**	32	**MNWTGLYTLLSGVNRHSTAIG**	**KRHKVHISGTLWWTYV**ISVVFRLLFE	VAEV**VYLIIRACARRAQRRS**NPP	liver [30], brain [31] testis [32]
				EINKLLSEQDGSLKDILRRS	
**γ**	33		EHGNRK**MRGRLLLTYMASIFFK**SVFE		
**γ**	34.7		KSSKV**RRQEGISRFYI**IQVVFRNALE	WRKIK**AAIRGVQARR**KSICEIRKKD	perch retina [33]
**γ**	35		TKSKMR**RQEGISRFYIIQVV**FRNALE	**WRKIKTAVRGVQARRK**SIYEIRNKD	perch retina [33]
**γ**	36		ARSKLR**RQEGISRFYIIQVV**FRNALE	WRKIKLAVR**GAQAKRK**SVYEIRNKD	pancreas, brain [34]
**γ**	37		EDG**RLRIRGALMGTYVISVL**CKSVLE		endothelium [35]
**γ**	38				xenopus oocyte [36]
**γ**	40		E**KAELSCWKEVNGKIVLQGTL**LNT	L**AELYHLGWKKIRQRLAKSR**Q	endothelium, heart [37]
**α**	43		EHGKVK**MRGGLLRTYIISIL**FKSVFE		cardiac [38]
**α**	44		DRG**KVRIAGALLRTYVFNII**FKTLFE		lens [39]
**α**	45		DGRRR**IREDGLMKIYVLQLLA**RTVFE		oligodendrocyte [40]
**α**	46		DRG**KVRIAGALLRTYVFNII**FKTLFE		lens [41]
**α**	50		GTKKF**RLEGTLLRTYVCHII**FKTLFE		lens [42]
**α**	56		ERGRI**RMGGALLRTYIFNII**FKTLFE		lens [43]
**α**	57		KIHKVP**LKGCLLRTYVLHIL****TRS**VLE	**RISLLQANNKQQVIRVNIPR**SK	retina [21]

CaM binding sequences in selected connexins. Green and red: predicted; Red: previously identified; Black: score 3 or <3; Green: score >3, <9; bold **green ** and **red**: score = 9; Purple: reported binding site [44], not predicted by algorithm (score 0), Underlined are the isoforms studied and the actual CL2 peptide sequences.

## 2. Results

### 2.1. Fluorescent CaM Derivatives to Investigate Ca^2+^-Dependent and -Independent Interactions with Cx CL2 Peptides

We used two fluorescently labelled CaM derivatives: TA-CaM and DA-CaM. TA-CaM is specifically labelled at Lys_75_, with the bright, environmentally sensitive TA-fluorophore, the fluorescence intensity of which increases as polarity is lowered [45]. TA-CaM fluorescence is increased by Ca^2+^ binding due to the exposure of hydrophobic pockets in the N- and C-lobes of CaM and changes further when a target domain is bound. We used TA-CaM to determine the Ca^2+^ dependence of the interaction with the Cx32, Cx35, Cx43, Cx45 and Cx57 CL2 peptides to take advantage of the Ca^2+^-induced fluorescence enhancement. Moreover, Ca^2+^/TA-CaM was most suitable for measuring the high affinity binding of Cx CL2 peptides because, due to its high brightness, it could be used at 10 nM concentration. 

DA-CaM, a FRET-based derivative ofthe T34C/T110C variant labelled with donor fluorophore AEDANS and quencher acceptor DDP, reports the distance and distance changes between the N- and C-lobes of CaM [46]. In the apo form, the two lobes are separated by 4 nm. Upon Ca^2+^ and target domain binding, the distance between the C- and N-lobe is typically shortened indicating the compaction of CaM structure as the N- and C-lobes lobes wrap around the peptide target stabilising its structure in an α-helix. Compaction of DA-CaM is indicated by greater quenching of the fluorescence of the donor AEDANS. DA-CaM fluorescence is not sensitive to Ca^2+^ binding, and hence, was suitable for studying Ca^2+^-independent interactions with Cx CL2 peptides. DA-CaM fluorescence changes reflecting its structural changes were thus utilised for both the Ca^2+^-dependent and -independent measurements of the association and dissociation kinetics of interactions with Cx CL2 peptides. 

### 2.2. High Affinity Binding of Cx32, Cx35, Cx43, Cx45 and Cx57 CL2 Peptides to Ca^2+^/TA-CaM

The fluorescence intensity of TA-CaM is dependent on the polarity of the fluorophore’s environment and it is affected both by Ca^2+^ and target protein or peptide binding [45]. Moreover, the high fluorescence brightness of TA-CaM allows measurement of high affinity target binding (nM *K*_d_) [45]. Ca^2+^/TA-CaM was titrated with the Cx CL2 peptides; *K*_d_ values were determined from normalised fluorescence titration curves. Four of the above Cx CL2 peptides were found to bind Ca^2+^/TA-CaM, reducing its fluorescence by between 20 and 50%. In contrast, the Cx43 CL2 peptide bound to Ca^2+^/TA-CaM with a 20% increase in fluorescence emission. The measured *K*_d_ values were in the range from ~20 to 150 nM, indicating high affinity interaction of the Cx CL2 peptides with Ca^2+^/CaM (Figure 2A, Table 2).

### 2.3. Conformation of Ca^2+^/CaM in Complex with Cx CL2 Peptides

As illustrated in Figure 2B, Ca^2+^/CaM binds to target peptide domains by embracing them with the two Ca^2+^ binding lobes and the connecting flexible loop can then adjust to different distances depending on where the residues of strong hydrophobic interaction are in the binding motif sequence. DA-CaM is a FRET derivative of CaM that contains a fluorescent donor at the periphery of one lobe and an acceptor molecule at the other. Donor quenching indicates FRET and decreased distance between the donor and acceptor. Wrapping around the smooth muscle MLCK derived Trp peptide, Ca^2+^/CaM forms a compact structure with maximum donor quenching of ~82% [46] With Cx NT and CT peptides less than full quenching has been found, indicating that the target peptide motifs are further apart. Equilibrium binding studies showed that Cx CL2 peptides typically quench Ca^2+^/DA-CaM fluorescence by 56–70% (Table 2), indicating that the peptide is not fully folded into an a-helix in complex with Ca^2+^/CaM.

### 2.4. Association and Dissociation Kinetics of Ca^2+^/DA-CaM Interactions with Cx CL2 Peptides

Gap junction channels operate in tissues dominated by widely differing signalling and metabolic environments and their connexins have different Ca^2+^ sensitivities. It is therefore likely that the kinetics of their interactions are varied. The kinetics of Ca^2+^/DA-CaM binding to and release from Cx CL2 peptides were studied by fluorescence stopped-flow. Upon the binding of Cx CL2 peptides to Ca^2+^/DA-CaM fluorescence decayed exponentially, reflecting quenching of Ca^2+^/DA-CaM fluorescence (Figure 3A,D,G,J,M). Thus, Ca^2+^/DA-CaM fluorescence quenching corresponds to the compaction of the CaM structure and the exponential decay indicates a conformational change. The mechanism considered is therefore one in which the initial binding to Ca^2+^/DA-CaM occurs without a fluorescence change. This is followed by a conformational change in which Ca^2+^/DA-CaM structure becomes more compact (Figure 3P). Plotting the observed rates at different peptide concentrations is expected to show a hyperbolic dependence in which the conformational change is limiting at higher concentrations. However, if the rate of the conformational change is too fast to measure, then the appearance of the plot approximates linearity. For Cx32, Cx35, Cx43 and Cx57 CL2 peptides, hyperbolic plots were obtained (Figure 3B,E,H,N); however, they were linear for Cx45 peptide (Figure 3K). Data were fitted to the equation derived from the mechanism in which binding is followed by a conformational change—in this case, CaM compaction—and in which the fluorescence change, in this case, exponential decay, represents the conformational change (Figure 3Q). Best fit parameters, equilibrium constant for the binding step (*K*_1_) and both the forward and the reverse rate constants for the isomerisation step were obtained for Cx32, Cx35 and Cx57 CL2 (*k*_+2_ and *k*_−2_) (Figure 3B,E,H,N). With the limiting rate of the conformational change too fast to measure, *k*_+2_*K*_1_ and *k*_−2_ values were fitted for the Cx45 CL2 peptide (Figure 3K, Table 3). 

Dissociation kinetics of the peptides from their complexes with Ca^2+^/DA-CaM was measured by trapping dissociated peptide by a high excess of Ca^2+^/CaM. Dissociation occurred with double exponential fluorescence increases, indicating that further slow stabilisation occurs following the observed single exponential decay in the binding process (Figure 3C,F,I,L,O). Both the rate of compaction of Ca^2+^/DA-CaM in the binding of the different Cx CL2 peptides and the dissociation rates cover a wide range consistent with different response kinetics by connexin isoforms to Ca^2+^, tuned for the cell or tissue environment in which they operate (Table 3). 

### 2.5. Association and Dissociation Kinetics of Apo-CaM Interaction with Cx CL2 Peptides

We were interested to see if we can detect apo-CaM binding to Cx CL2 peptides using DA-CaM, the conformation of which is insensitive to [Ca^2+^] changes, but which indicates changes in the distance between the N- and C-lobe with high sensitivity. Evidence for binding came in the form of complex conformational changes in DA-CaM when interacting with the peptides. As seen in Figure 4A, a small fluorescence quenching (<10%) occurred up to ~10 μM peptide concentration in the association reaction of Cx32 CL2 peptide with apo-DA-CaM. However, at higher peptide concentrations, the fluorescence signal ‘flipped’ indicating stretching of CaM beyond its length when free. Slow dissociation of the Cx32 CL2 peptide–apo-DA-CaM complex indicates high affinity binding (Figure 4B). Similarly, small signal amplitudes, slow binding and ‘flipping’ are seen in the Ca^2+^-independent interactions of each Cx CL2 peptide studied together with slow dissociation rates, as evidence of binding to apo-CaM (Figure 4C–F). 

As CaM structure is more compact when the target domain folds into an α-helix, we hypothesised that helix-promoting TFE [47] might increase the rate and amplitude of apo-DA-CaM binding to Cx CL2 peptides and tested this hypothesis on the Cx43 and Cx57 CL2 peptides. As predicted, both the rate (400- and 600-fold, respectively) and the amplitude of quenching of DA-CaM fluorescence increased significantly in the presence of 15% TFE (Figure 4G,H and Table 4). Thus, apo-CaM binding does not stabilise the Cx CL2 peptides in an α-helical conformation.

### 2.6. Ca^2+^-Dependence of the Interaction of Cx32, Cx35, Cx43, Cx45 and Cx57 CL2 Peptides with Lys_75_-TA-CaM

Apo-DA-CaM binding to Cx CL2 peptides is consistent with CaM anchoring to gap junctions in resting cells. We were interested in determining if the interaction at low [Ca^2+^] actually occurs with apo-CaM or with partially (but tightly) Ca^2+^-bound CaM, as this would indicate by which mechanism anchoring may occur in physiological conditions. Thus, we investigated the Ca^2+^ dependence of the interaction between TA-CaM and the Cx CL2 peptides by equilibrium titration.

TA-CaM fluorescence responds to C-lobe Ca^2+^ binding, as indicated by the low [Ca^2+^] value of ~550 nM at half-maximal fluorescence change (defined as *K*_d_ for Ca^2+^) (Figure 5A), while *K*_d_ of CaM for Ca^2+^ is in the μM range [6]. Mixtures of TA-CaM and a Cx CL2 peptide were titrated with Ca^2+^ to see the effect of Cx CL2 peptide binding (Figure 5B–F). The binding curves were fitted to the Hill equation to obtain *K*_d_ and Hill coefficient values (*n*) (Table 2). TA-CaM fluorescence increases 5.5-fold by Ca^2+^ binding with *n* of 5.3 ± 0.3. In the presence of Cx CL2 peptides, the TA-CaM fluorescence increases were Cx32, 2-fold; Cx35, 3-fold; Cx43, 2-fold, Cx45, 2-fold and Cx57, 3-fold. 

Cx32 and Cx35 and Cx43 CL2 peptides caused a left shift in the Ca^2+^ dependency curve to *K*_d_-s of 311 nM, 392 nM and 340 nM, respectively, indicating Ca^2+^-dependent binding in which Ca^2+^ binding to CaM strengthens peptide binding and vice versa. The high affinity of peptide binding—but an only mildly increased affinity for Ca^2+^ of the C-lobe (Figure 5B–D)—implies that N-lobe Ca^2+^ binding is significantly increased in affinity by peptide binding. The lack of a change in TA-CaM fluorescence at resting [Ca^2+^] of ~50 nM taken together with evidence of binding in the absence of Ca^2+^ (see above) indicates Ca^2+^-independent apo-CaM binding to Cx32 and Cx35 and Cx43 CL2 in resting cells (Table 2).

Cx45 and Cx57 CL2 peptide binding evoked a left shift and independence of the two CaM lobes. The left shift to the presumed C-lobe Ca^2+^ binding to *K*_d1_ of 70 nM indicates that, at resting [Ca^2+^] concentrations (~50 nM), CaM anchoring to the Cx45 CL2 in a gap junction would be based on CaM with a very tightly bond Ca^2+^ rather than apo-CaM. The higher *K*_d2_ of 597 nM indicates N-lobe binding upon Ca^2+^ stimulation (Figure 5E). The same scenario is applicable to Cx57 with a 32 nM apparent lower *K*_d1_. The higher *K*_d2_ of 1.51 μM required for the N-lobe binding to Cx57 suggests that Cx57 operates in environments in which intracellular [Ca^2+^] rises to higher than the often seen 500 nM upon cell stimulation (Figure 5F).

### 2.7. Ca^2+^.CaM Block of Gap Junctions in HUVEC Is Prevented by Inhibition with mTrp Peptide

#### 2.7.1. HUVEC Express CX43 at Cell-Cell Contacts and Are Functionally Gap Junction Coupled

Endothelial cells express Cx43 gap junction proteins [23]. Figure 6 shows HUVEC co-stained for endogenous Cx43 and the endothelial specific cell-junction adhesion molecule, VE-cadherin to mark cell–cell boundaries. Cell–cell contacts were clearly defined by VE-Cadherin labelling and decorating these contact sites were numerous Cx43 positive puncta showing the presence of gap junction complexes. 

To determine if HUVEC were functionally coupled, we analysed the spread of extracellularly applied LY following scratch wounding as described in Methods. Spread of LY was seen on both sides of scratch wound and the extent of the spread increased with time after scratch wound formation (Figure 7), confirming the cells are functionally gap junction coupled. 

##### Disruption of Endothelial Gap Junction Coupling by PMA and 18 ßGRR

Having established that the HUVEC monolayer cultures are functionally gap junction coupled, we next tested two known gap junction uncouplers, 12-phorbol 13-myristate acetate (PMA, 100 nM) and 18-β-glycyrrhetinic acid (18ßGRR; 100 µM), to confirm that changes in gap-junctional coupling could be detected and quantified. Figure 8A shows images from one such experiment and is representative of four separate independent experiments. Pre-treatment of HUVEC for 60 min with either PMA or 18ßGRR significantly reduced LY spread indicating inhibition of gap junction coupling. No evidence for decreased cell viability following drug treatments was observed. 

##### Ca^2+^-Dependent Gap Junction (GJ) Uncoupling

Previous studies have shown that CaM causes a Ca^2+^-dependent closure of Cx43 GJs in cells exposed to the Ca^2+^ ionophore ionomycin [12,15]. Exposure of fura-2 loaded HUVEC to ionomycin (2 µM) resulted in an initial fast increase in [Ca^2+^]_i_ that declined to resting levels in the absence of external Ca^2+^ (*n* = 11 cells) or to a lower but maintained plateau elevation in the presence of external Ca^2+^ (*n* = 9 cells) (Figure 9A). Pre-treatment (8 min) of HUVEC with ionomycin in the presence of external Ca^2+^ significantly reduced the spread of LY following scratch formation (Figure 9B,C). The effect was greatly reduced if external Ca^2+^ was omitted during ionomycin stimulation. 

##### mTrp Peptide Reverses Ca^2+^-Dependent GJ Uncoupling in HUVEC

To determine whether the Ca^2+^-dependent GJ uncoupling observed in HUVEC following ionomycin treatment is mediated by CaM, we incubated cells with the membrane permeant CaM inhibitory peptide mTrp (7–17.6 µM) or an inactive control peptide at the same concentrations (Figure 10). Peptides were included during vehicle control or ionomycin pre-treatment of cells (8 min) and during assay of LY spread (8 min) in Ca^2+^-containing media. In the absence of ionomycin stimulation there was no significant difference in LY spread between control peptide or mTrp at the concentrations tested (7 µM; *n* = 23 observations for each condition pooled from 2 independent experiments, *p* = 0.0953, 17.9 µM *n* = 44 and *n* = 48 observations for each condition pooled from 2 independent experiments, *p* = 0.65, ANOVA GraphPad Prism). In control, peptide treated cells ionomycin pre-treatment significantly reduced LY spread following scratch wounding (*p* < 0.0001). However, treatment with mTrp reduced the inhibition of LY spread compared to the control peptide in a dose-dependent fashion (7 μM; *p* = 0.0073, 17.9 μM *p* = 0.0001).

## 3. Discussion

We demonstrated high affinity (*K*_d_ in low 10 s of nM) Ca^2+^/CaM binding to the Cx CL2 peptides of Cx32, Cx35, Cx43, Cx45 and Cx57. In structural terms, our data show partial collapse of Ca^2+^/CaM around the peptides. As illustrated in Figure 2B, Ca^2+^/CaM binds to target peptide domains by embracing them with the two Ca^2+^ binding lobes, the connecting flexible loop can adjust to different distances depending on where the residues of strong hydrophobic interaction are in binding motif sequence. Our data show complete collapse of Ca^2+^/CaM when binding to the smooth muscle (sm) MLCK derived Trp and mTrp peptide, in agreement with NMR and crystal structures [49,50]. Wrapping around the sm MLCK derived Trp and mTrp peptides, the quenching of the donor is ~82% ([45] and Table 2). With Cx CL2 peptides, less quenching—typically ~40%—has been found [51], similar to that previously described for Cx32 NT and CT peptides [7], indicating that the CaM binding motifs of Cx peptides are further apart. This can occur either because the peptide is not fully folded into an α-helix, or because the binding residues are not in the 1-5-10, but, for example, in the 1–14 configuration. For Cx CL2 peptides, this is not surprising given the disruption of helical folding by one or two Gly residues in their sequence, also indicated by secondary structure prediction of the Cx CL2 peptides (Figure 11).

Myllykoski et al. reported complete CaM collapse in complex with a shorter, 15-residue Cx43 CL2 peptide [52] based on small angle X-ray scattering and binding energetics. Interestingly, their measured affinity is also in the μM rather than nM range, similar to that measured by Zhou et al. for a Cx43 CL2 peptide extended eight residues N-terminally [53]. Incidentally, in the same paper Zhou et al. incorrectly cited the measurement of a 1.2 μM *K*_d_ for an N-terminal (NT) peptide from Cx43, when in the cited paper an NT CaM binding domain of Cx32 was identified with *K*_d_ of 27 nM for TA-CAM [6]. Cx43 has no predicted NT CaM binding domain. Cx32 has been reported to have a CT CaM binding domain [44], designated here as Cx32 CT peptide (see Materials and Methods). This peptide represents a different section and sequence from those previously identified [6,7]. We were unable to detect any signals indicating CaM binding to this peptide. 

In our CL2 peptide design, an additional 6 amino acids overlapping the N terminal section of trans-membrane helix 3 (TM3) were included. For our Cx43 CL2 peptide, we report a *K*_d_ of 18 nM for Ca^2+^/TA-CaM binding and partial CaM collapse in complex with the peptide. Thus, we hypothesise that the Ca^2+^/CaM block may involve a structural rearrangement in which TM3 is ‘pulled out’ of the membrane by the bound Ca^2+^/CaM. The process appears to take about 8 min and it is not inconceivable that inhibition develops slowly while more than 1 Ca^2+^/CaM molecule associates with a gap junction hexamer. Slow dissociation of Ca^2+^-independently bound CaM may also contribute to limiting its replacement by Ca^2+^/CaM.

In DA-CaM, we found a tool for investigating Ca^2+^-independent target binding. In effect, the CaM structural changes associated with the collapse of its structure when binding to Cx CL2 peptides produced very small signals, ≤10%. As these interactions are based on charge rather than hydrophobic interactions, the lack of a large structural collapse is not surprising. The best evidence we obtained on Ca^2+^ independent CaM binding to Cx CL2 peptides came from their slow dissociation kinetics (Figure 4). The small signals made equilibrium titrations unfeasible. Moreover, we observed a ‘flipping’ of the DA-CaM structure: as the peptide concentration increased, DA-CaM underwent stretching rather than collapsing. We hypothesise that underlying process of DA-CaM structural ‘flipping’ may be a helix-to-coil transition of the peptide structure [55]. Secondary structure prediction suggests a tendency to form an α-helix in the C-terminal half of the peptides, as opposed to existing as a random coil. We hypothesise that a helix-to-coil transition occurring at higher concentrations of the peptides would force apo-DA-CaM to adjust to a more extended conformation compared to when it binds to helically folded peptides.

mTrp peptide interacts with apo-DA-CaM in a similar fashion to Cx CL2 peptides. With the association of mTrp and apo-DA-CaM, there is a clear, biphasic fluorescence change: initial rapid quenching is followed by a second isomerisation in which DA-CaM partially stretches back. mTrp dissociation from apo-DA-CaM occurs in two phases: a fast of 130 s^−1^ and a slow phase of 0.9 s^−1^, in both the fluorescence increases, indicating that the reverse pathway is different from the on-pathway. Rapid dissociation of the apo-DA-CaM complex is consistent with specific—but lower—affinity interaction compared to Cx CL2 peptides, as expected for a peptide derived from smooth muscle (sm) MLCK, an enzyme highly Ca^2+^-dependent, regulated by CaM (Appendix A). Ca^2+^ dependence of TA-CaM fluorescence in the presence of mTrp peptide shows significant binding at 50–100 nM [Ca^2+^], consistent with previously proposed C-lobe anchoring of CaM to sm MLCK in resting cells [56] (Appendix A, Table 2). 

We used the helix-inducing TFE, predicting that it would facilitate apo-CaM interaction, manifesting in a faster rate of association and leading to a more compact conformation in the resulting complex. As shown in Figure 4G,H, in 15% TFE, in binding to the Cx57 and Cx43 CL2 peptides, both the rates of the DA-CaM conformational change and the amplitude of DA-CaM fluorescence quenching—indicating CaM structural collapse—are ~3-fold increased. 

TA-CaM fluorescence increases in response to Ca^2+^-binding induced exposure of hydrophobic pocket(s). When we see no change in TA-CaM fluorescence at resting [Ca^2+^] levels ~50 nM, we conclude that no such exposure occurs which, taken together with evidence from the kinetic studies of apo-CaM binding, is consistent with Ca^2+^-independent CaM binding in resting cells, likely by the C-lobe. This scenario applies to Cx32, Cx35 and Cx43.

Cx45 and Cx57 CL2 peptide binding to TA-CaM involved a high affinity Ca^2+^ site, with apparent *K*_d_-s of 75 and 32 nM, respectively (Table 2). Thus, anchoring of CaM to Cx45 and Cx57 gap junction Cx-s in resting cells would occur in a partially Ca^2+^-bound complex. This distinction is important as the binding modes of apo- and Ca^2+^-CaM are distinct. 

Previous studies have shown that CaM causes a Ca^2+^-dependent closure of Cx43 GJs in cells exposed to the Ca^2+^ ionophore ionomycin [12,15]. Here, we utilised the high affinity, membrane permeant mTrp peptide to release the Ca^2+^/CaM block of gap junction coupling, as well as the mCtrl peptide in which two critical hydrophobic residues were substituted by Glu, to show the specificity of the mTrp peptide. Several connexin isoforms have multiple binding sites for Ca^2+^.CaM; however, the common denominator appears to be predicted binding site in the second half of the intracellular loop (CL2) that is present in almost all connexins (see Table 1 and references within). We used HUVEC, which predominantly express Cx43, with a number of other isoforms—Cx32, Cx37 and Cx40 also expressed—to see what fraction of gap junction permeability can be blocked by Ca^2+^.CaM. Using the scratch assay as described in Methods, we found that unstimulated HUVEC had unobstructed dye spread which was completely blocked following Ca^2+^ influx evoked by ionomycin stimulation. Administration of the mTrp peptide, which has a 15 pM affinity for Ca^2+^.CaM, prevented the Ca^2+^-induced block, indicating that the gap junction blockade was mediated by Ca^2+^.CaM interaction with the various connexin isoforms present in HUVEC, and furthermore, that Ca^2+^/CaM binding is sufficient for blocking gap junction coupling. Of the expressed isoforms, Cx37 and the dominant Cx43 isoforms only have CL2 as a Ca^2+^/CaM binding domain. Ca^2+^/CaM inhibition in HUVEC thus supports the suggestion by Zhou et al. [53] that Ca^2+^/CaM–Cx CL2 domain interactions may be sufficient for blocking gap junctions. Notably, the release of the Ca^2+^/CaM block was a slow process, consistent with our proposal that major conformational changes, possibly involving TM3 may underlie CaM and Ca^2+^/CaM interactions with gap junction Cx-s. In summary, we have demonstrated inhibition of gap junction coupling by using the very specific and high affinity mTrp peptide as inhibitor (and mCTRL as control) in HUVEC that expresses a number of connexin isoforms, some of which only have CL2 as calmodulin binding domain. Furthermore, we demonstrated the concentration and time dependence of inhibition, thereby establishing an assay that can be used in the future—for example, for comparing the ability of endogenous calmodulin variants to block gap junction coupling. 

In conclusion, both Ca^2+^-dependent and -independent binding between Cx CL2 peptides and CaM appear to be of high affinity, albeit via very different molecular interactions. We propose a model in which the C-terminal, highly hydrophobic section of the CL2 domain of the intracellular loop remains buried in the membrane in resting cells, with the CaM C-terminal domain anchored to the N terminal part of the CL2 domain. Ca^2+^ binding to CaM allows the N-lobe to capture the C-terminal peptide binding site by pulling a stretch of TM3 residues out of the membrane and thereby allowing blocking of the gap junction pore. Ca^2+^ dissociation releases the TM3 hydrophobic stretch of CL2 which slides back into the membrane, resulting in the gap junction pore opening.

## 4. Materials and Methods

### 4.1. Peptides and Proteins

Peptides were obtained from Genosphore Biotechnologies, Boulogne-Billancourt, France. They had >95% purity and their sizes were analysed by mass spectrometry. The amino acid sequences of the peptides are: Cx32 CL2 peptide: KRHKVHISGTLWWTYVISVV; Cx32 CT peptide: EINKLLSEQDGSLKDILRRS [44]; Cx35 CL2 peptide: TKSKMRRQEGISRFYIIQVV; Cx43 CL2 peptide: KVKMRGGLLRTYIISILFK; Cx45 CL2 peptide: RRRIREDGLMKIYVLQLLAR; Cx57 CL2 peptide: KIHKVPLKGCLLRT-YVLHIL; Myr-Trp peptide (mTrp): N-[myr]-RRKWQKTGHAVRAIGRL-amide and myr-Ctrl peptide: N-[myr]-RRKEQKTGHAVRAIGRE-amide [24].

Peptide concentrations were determined spectroscopically using extinction coefficients calculated by Protein calculator v3.4: for Cx32 CL2 peptide, *ε*_0(280nm)_ = 12,660 M^−1^cm^−1^; for Cx35, Cx43, Cx45, and Cx57 CL2 peptides, *ε*_0(278nm)_ = 1280 M^−1^cm^−1^ and for mTrp peptide, *ε*_0(280nm)_ = 5700 M^−1^cm^−1^. Myr-Ctrl peptide and Cx32 CT peptide concentrations were determined by weight.

Calmodulin and T34C/T110C-calmodulin were expressed in *E. coli*, purified and double-labelled DA-CaM [45] and Lys_75_-labelled TA-CaM [46] were generated as previously described.

### 4.2. Ca^2+^-TA-CaM Equilibrium Titration with Cx CL2 Peptides

The titrations were performed using a Flurolog-3 fluorimeter (Horiba UK, Northampton, UK). The sample was excited at 365 nm and the fluorescence emission was recorded at 415 nm. TA-CaM (10 nM) in 50 mM HEPES-K^+^, 100 mM KCl, 2 mM MgCl_2_ pH 7.5 (assay buffer) and 100 µM CaCl_2_, at 20 °C was titrated stepwise with 600 nM and 6 µM stock solutions of Cx CL2 peptides in the same buffer until no further changes in fluorescence emission intensity were observed. Experiments were done in triplicates and the normalised fluorescence intensities were averaged. For analysis, the fluorescence change was inverted and the data for Ca^2+^-dependent binding were fitted to the equation for one site-specific binding curve (y = *B*_max_∗x/(*K*_d_ + x)), where x corresponds to [Cx CL2 peptide] and *B*_max_ represents the maximal signal (max fraction bound = 1), using GraphPad Prism 9.

### 4.3. Kinetic Measurements

The kinetics of Ca^2+^/DA-CaM and DA-CaM interactions with Cx CL2 peptides was measured using a TgK Scientific KinetAsyst™ double-mixing stopped-flow apparatus (TgK Scientific Ltd., Bradford-on-Avon, UK) in single mixing mode. The fluorescence was excited at 340 nm and the emission was detected by a photomultiplier tube with a long pass filter (>400 nm). For Ca^2+^-dependent association, 500 nM DA-CaM and different concentrations of a Cx CL2 or mTrp peptide were rapidly mixed in assay buffer and 1 mM CaCl_2_. For Ca^2+^-dependent dissociation, pre-mixed 500 nM DA-CaM with Cx CL2 peptide at saturating concentration (500 nM Cx32, 2 µM Cx35, 1.5 µM Cx43, 1 µM Cx45 or 2 µM Cx57 CL2 peptide) or 0.5 µM mTrp peptide in assay buffer and 1 mM CaCl_2_ was rapidly mixed with 20 µM CaM. For Ca^2+^-independent association, 500 nM DA-CaM in assay buffer containing 5 mM EGTA at 20 °C was rapidly mixed with different concentrations of a Cx CL2 or mTrp peptide. For Ca^2+^-independent association in the presence of trifluoroethanol (TFE), 500 nM DA-CaM in assay buffer containing 5 mM EGTA and 15% TFE was rapidly mixed with 5 µM or 17.5 µM Cx43 and 5 µM or 30 µM Cx57. For Ca^2+^-independent dissociation, 500 nM DA-CaM with Cx CL2 peptide (10 µM Cx32, 25 µM Cx35, 15 µM Cx43, 20 µM Cx45 or 25 µM Cx57 CL2 peptide) or 10 µM mTrp peptide in assay buffer containing 5 mM EGTA was rapidly mixed with 300 µM CaM. All concentrations given are those in the mixing chamber. Data shown are averages of 3 to 5 repeats. All experiments were carried out at 20 °C. Data were analysed using Kinetic Studio software (TgK Scientific Ltd., Bradford-on-Avon, UK). 

### 4.4. TA-CaM Ca^2+^ Titration in the Presence of Cx CL2 Peptides

The apparent dissociation constant (*K*_d_, the [Ca^2+^] concentration at which half-maximum fluorescence intensity is achieved) of TA-CaM in the presence of the different Cx CL2 peptides was determined by following the change in fluorescence emission (λ_ex_ = 365 nm, slit 1 nm; λ_em_ = 415 nm, slit 15 nm). The mixture of 10 nM TA-CaM and 600 nM Cx32, 450 nM Cx35, 250 nM Cx43, 1.1 µM Cx45 or 900 nM Cx57 CL2 and 30 nM mTrp peptide in assay buffer and 5 mM EGTA were titrated with 325 mM Ca^2+^ in assay buffer using a syringe pump (10 µL/min) while constantly stirring. The fluorescence was corrected for dilution and [Ca^2+^] concentration was calculated using Maxchelator software. The data were fitted to the Hill equation with specific binding (y = *B*_max_∗x^n^/(*K*_d_^n^ + x^n^)) using GraphPad Prism 9.

### 4.5. Cell Culture

Cryopreserved pooled primary HUVEC were purchased from Lonza (Valais, Switzerland). HUVEC were grown in a human growth medium (HGM) which comprised media M199 supplemented with 20% FCS. The antibiotic gentamicin (50 µg/mL) along with ECGS (endothelial cell growth supplement; 30 µg/mL) and heparin (10 U/mL) were added to the media. HUVEC were grown on 24 well trays or 9 mm glass coverslips that had been pre-treated with porcine skin gelatine (1% *w*/*v*) for 30 min at 37 °C. Cells were cultured for 4 days in a 5% CO_2_ incubator at 37 °C until fully confluent as assessed by visual examination by light microscopy.

### 4.6. Immunocytochemistry

For analysis of Cx43, VE-cadherin localisation and expression in HUVEC, we used a pre-validated mouse anti-Cx43 monoclonal antibody directed to a C-terminal epitope (clone P4G9, EMD MilliporeSigma, Burlington, MA, USA) at a 1:25 dilution and a rabbit polyclonal anti-VE-cadherin antibody (D87F2, Cell signalling, Leiden, The Netherlands) at 1:400 dilution. HUVEC grown on 9 mm porcine gelatine coated coverslips were first washed carefully with phosphate buffered saline (PBS) and then placed into a solution of 3% paraformaldehyde (PFA) in PBS for 15 min. The coverslips were then washed into PBS supplemented with 50 mM NH_4_Cl and incubated at room temperature (RT) for 15 min. The NH_4_Cl solution was then removed and replaced with PBS, supplemented with the detergent TX-100 (30 µL/10 mL) for 5 min at RT. The coverslips were then washed into a phosphate gelatine and saponin solution (PGAS) that contained (*w*/*v*): 0.2% gelatin, 0.02% saponin and 0.02% NaN_3_ in PBS and stored at 4 °C until processed for immunocytochemistry. For immunolabelling, a 40 μL droplet of PGAS solution containing single or multiple primary antibodies at the dilutions described above was placed on a sheet of parafilm within a humidified chamber and a 9 mm coverslip of fixed and permeabilised HUVEC placed cell side down on the droplet for 60 min at RT. The coverslips were then transferred to a well of a 24-well plate and washed 3 times with PGAS before being removed and placed cell-side down on a 40 µL droplet of PGAS containing species appropriate secondary antibodies coupled to FITC, RedX or Cy5 (Jackson ImmunoResearch Europe Ltd., Ely, UK). Coverslips were then incubated for 60 min at RT in the dark before being washed 3 times in PGAS, followed by 3 washes in PBS and then mounted cell-side down in a 5 µL droplet of Mowiol-488 on a glass microscope slide. Mounted coverslips were kept in the dark at RT for 24 h before viewing on a confocal microscope. 

### 4.7. Confocal Microscopy

Fixed and immune-labelled HUVEC were imaged by confocal microscopy (Bio-Rad Radiance 2100 (Bio-Rad, Hercules, CA, USA)) using a 60× 1.4 NA oil immersion objective. Light for FITC, RedX and Cy5 excitation were 488 nm (50 mW Kr/Ar laser), 542 nm (green laser diode) and 637 nm (5 mW red laser diode), respectively. Sequential image capture was used for multi-labelled samples and images were acquired using the same laser settings at each wavelength using a Kalman filter (4 frames). Images (1024 × 1024 pixels) were acquired at zooms 1.4 or 2.6. Data was recorded in BioRad pic format and converted to tif format in ImageJ. Figures were made in Photoshop CC (Adobe).

### 4.8. Epifluorescence Imaging of Changes in [Ca^2+^]_i_

HUVEC were grown on 24 mm square uncoated glass coverslips in growth medium. For Fura-2 loading, the cells were transferred into Hanks solution (140 mM NaCl, 5 mM KCl, 1.8 mM CaCl_2_, 2 mM MgCl_2_, 10 mM glucose, 10 mM HEPES-NaOH pH 7.4) supplemented with 2.5 µM Fura-2AM (Molecular Probes, Eugene, Oregon, USA) diluted from a stock solution of 5 mM Fura-2AM in DMSO with 2.0% Pluoronic F-127. Cells were incubated at room temperature in the dark for 20 min before being transferred into fresh Hanks solution and further maintained in the dark at RT until use. Cells were transferred to the stage of an Olympus IX71 inverted fluorescence microscope and maintained at 32–34 °C. The imaging system comprised an excitation monochromator (Optoscan, Cairn Research), U-Plan-Apo ×100 1.35 NA objective and an Ixon EMCCD camera (Andor). Fura-2 was illuminated sequentially with 355 ± 5 nm and 380 ± 5 nm light and wavelength switching synchronised with image capture using WinFluor software (Dr John Dempster, Strathclyde University). The EMCCD camera was operated in frame transfer mode at full gain and cooled to −70 °C. Full frame (512 by 512 pixels) images were acquired at 30 frames/s. Ionomycin was applied by droplet application.

### 4.9. Multiparametric Scrape Loading-Dye Transfer Assay

The assay is described in detail in [48]; it is an adaption of the traditional fluorescence microscopy-based scrape loading dye transfer (sl-dt) technique. Briefly, HUVEC in 24 well trays were rinsed into +Ca+Mg-PBS. The solution was then replaced with fresh +Ca+Mg-PBS supplemented with Lucifer Yellow (LY; 1 mg/mL), propidium iodide (PI; 15 μM) and Hoechst 33,342 (90 μM) (dye transfer buffer). Immediately after addition of the dye transfer buffer, 3 parallel cuts in the cell monolayer in the central part of each well in the 24 wells were made using a micro scalpel knife with a flat edge stainless steel blade. The cells were then incubated at 37 °C for 8 or 15 min, as specified, before being washed twice with +Ca+Mg-PBS to remove the dye transfer buffer and then fixed in 3% PFA in PBS. Fresh dye transfer buffer was used for each experiment and the plates were stored in the dark at 4 °C.

### 4.10. Fluorescence Imaging for Dye Transfer Assay

For quantitative analysis of dye transfer and cell viability, images were acquired from 2 to 3 regions within each cut (6–9 measurements per well) using an EVOS M5000 microscope (Invitrogen, Waltham, MA, USA) equipped with a 10 × 0.3 NA air objective and EVOS LED light cubes for bright field imaging, FITC, RedX and DAPI. Images were acquired sequentially and exported to ImageJ for processing. 

### 4.11. Treatment of Cells with Gap Junction Uncoupling Agents and CaM Inhibitory Peptides

Phorbol 12-myristate 13-acetate (PMA) or 18-β-glycyrrhetinic acid (18BGRR) was made up in dimethysulphoxide (DMSO) to a 10 mM or 50 mM stock, respectively, and diluted into HGM to a final concentration of 100 nM or 100 µM, respectively. Cells were incubated in PMA or 18BGRR acid containing solution for 60 min at 37 °C. The PMA or 18BGRR solution was then carefully removed and the cells were washed once in +Ca+Mg-PBS. The PBS was then replaced with the dye transfer buffer (see above) and immediately 3 parallel cuts were made in the cell monolayer in the central part of each well using a micro scalpel knife with a flat edge stainless steel blade. For each experiment drug treatments were carried out in triplicate wells each with 3 cuts per well. Cells were then processed as described above. For analysis of the Ca^2+^ dependency of gap junction coupling, HUVEC were incubated in either +Ca+Mg-PBS (+Ca) or -Ca+Mg +0.2 mM EGTA (-Ca) containing PBS and stimulated with vehicle (0.001% DMSO), or 2 μM ionomycin for 8 min. Cells were then transferred to dye transfer buffer made using +Ca or −Ca PBS, supplemented with vehicle or 2 μM ionomycin, and 3 parallel cuts were made in cell monolayers in each well. Cells were incubated for a further 8 min prior to fixation. The cell permeant CaM inhibitory peptide, mTrp, and the control peptide mCtrl are described in [24]. A stock solution of 3.6 mM was made to generate working concentrations of 6–18 µM. Peptides were kept on ice till needed and cells preincubated for 8 min with mCtrl, or mTrp peptides in the presence or absence of 2 μM ionomycin, before being transferred to the dye transfer buffer supplemented with the peptides and vehicle, or 2 μM ionomycin, and incubated for a further 8 min prior to fixation and processing. 

### 4.12. Data Analysis

Data analysis was carried out in the open-source-based software program Fiji ImageJ. A custom-made macro (Cut_Analyzer.ijm) published in the Appendix A to [48] was used for the automated quantitation of gap junction intercellular communication using images acquired for the LY signal. Analysis of total cell number (Hoechst channel) and cell viability (propidium iodide channel) was made using custom made macros ([48] and Appendix A).

### 4.13. Statistical Analysis

Data are presented as mean ± s.e.m., unless otherwise stated. Statistical analysis of the effect of gap junction blockers and calmodulin inhibitory peptides on dye transfer, cell viability or cell count was carried out using ANOVA or *t*-tests, as appropriate in GraphPad Prism.

## Figures and Tables

**Figure 1 ijms-24-04153-f001:**
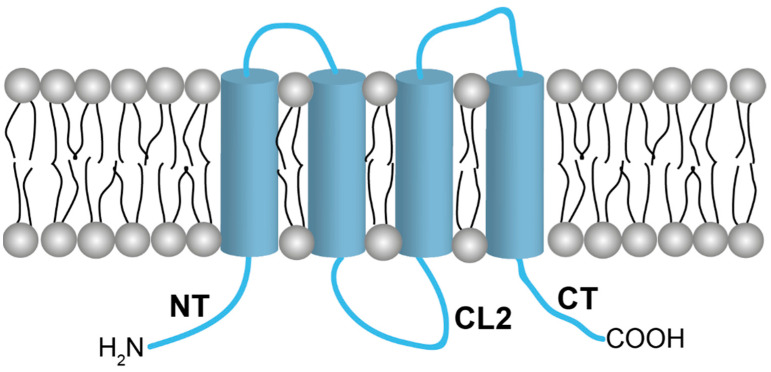
Topological schematic of a gap junction Cx.

**Figure 2 ijms-24-04153-f002:**
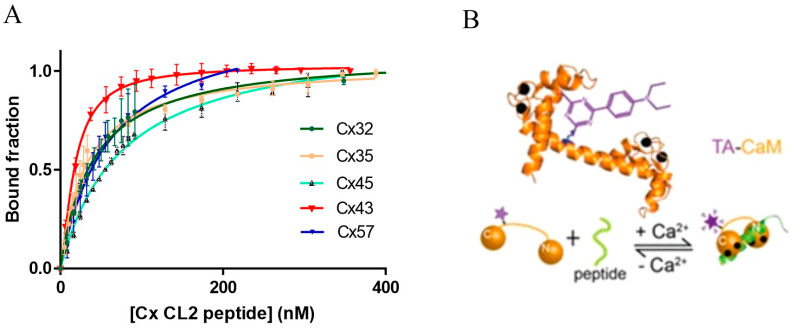
Ca^2+^-dependent CaM binding of Cx CL2 peptides. (**A**) Equilibrium binding of 10 nM Ca^2+^/TA-CaM to Cx CL2 peptides in 50 mM HEPES-K^+^, 100 mM KCl, 2 mM MgCl_2_ and 100 µM CaCl_2_, pH 7.5 at 20 °C. The fluorescence intensity of Ca^2+^/TA-CaM derivatives decreased upon peptide addition. The data were corrected for dilution, inverted, normalised and the bound fraction is plotted against the Cx CL2 peptide concentration. The error bars represent S.D. (**B**) Schematic drawing of TA-CaM, TA-Cl is covalently bound to Lys_75_ [45]. Binding of Ca^2+^ to TA-CaM in the presence of peptide leads to a fluorescence emission increase of the fluorophore (purple star).

**Figure 3 ijms-24-04153-f003:**
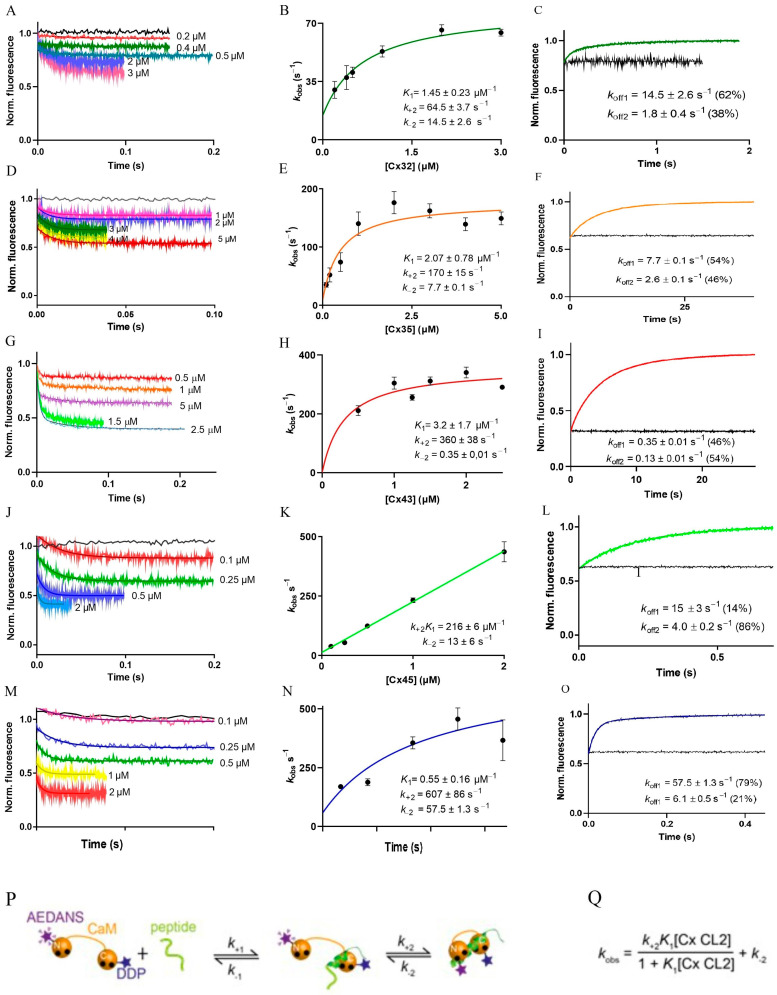
Association and dissociation kinetics of Ca^2+^/DA-CaM and (**A**–**C**) Cx32 CL2, (**D**–**F**) Cx35 CL2, (**G**–**I**) Cx43 CL2, (**J**–**L**) Cx45 CL2 and (**M**–**O**) Cx57 CL2 peptides. (**A**,**D**,**G**,**J**,**M**) Cx CL2 peptide concentration dependent association kinetic records with the corresponding mono-exponential fits. The trace at 1 represents Ca^2+^/DA-CaM mixed with buffer and the trace at 0 represents buffer mixed with buffer. (**B**,**E**,**H**,**K**,**N**) Observed rates (*k*_obs_) obtained from the mono-exponential fit plotted against the Cx CL2 peptide concentration. The error bars represent SD. The data were fitted using the equation (**Q**) derived from the model shown in scheme (**P**). Fitted parameters are stated in each panel. (**C**,**F**,**I**,**L**,**O**) Dissociation kinetic records of Ca^2+^/DA-CaM/Cx CL2 peptide solutions mixed with excess CaM (20 μM) and their single exponential fit. The line at 0 represent the record for buffer mixed with buffer, while the straight line at the top is the recording of Ca^2+^/DA-CaM/Cx CL2 peptide solution mixed with buffer only. (**P**) Kinetic scheme of rapid binding followed by a conformational change in which Ca^2+^/DA-CaM becomes more compact. (**Q**) Equation derived for fitting *k*_obs_ values to Scheme in Panel **P**.

**Figure 4 ijms-24-04153-f004:**
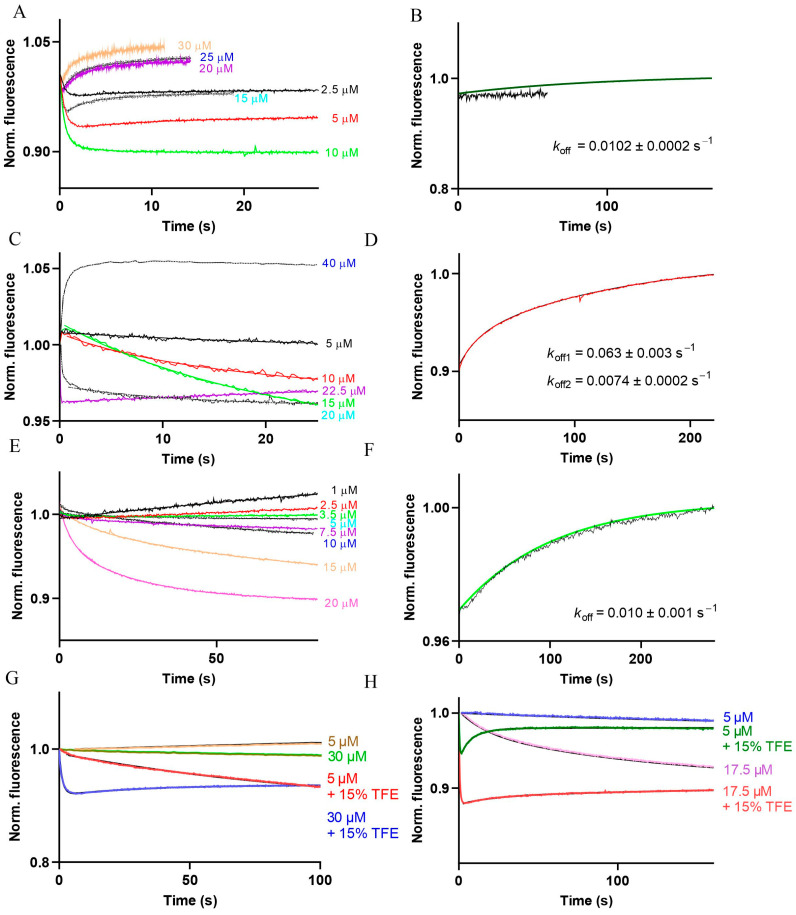
Association and dissociation kinetics of apo-DA-CaM and (**A**,**B**) Cx32 CL2, (**C**,**D**) Cx43 CL2, (**E**,**F**) Cx45 CL2, (**G**) Cx57 CL2 (with and without 15% TFE) and (**H**) Cx43 CL2 (with and without 15% TFE) peptides, respectively.

**Figure 5 ijms-24-04153-f005:**
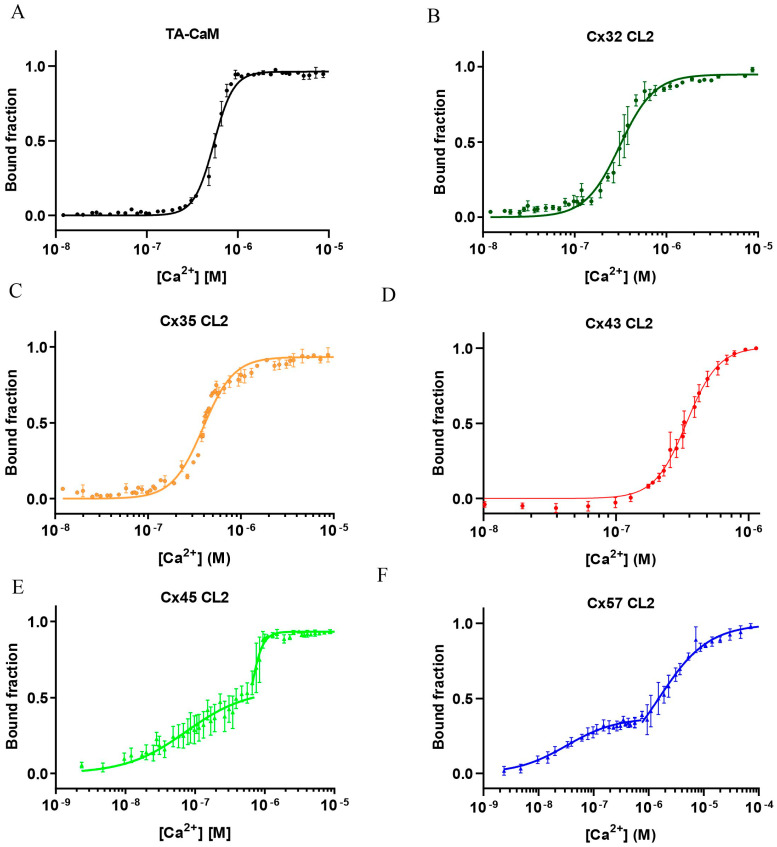
Ca^2+^ equilibrium binding of TA-CaM in the presence of Cx CL2 peptides. The bound fraction is plotted against the [Ca^2+^] concentration. (**A**) TA-CaM; TA-CaM in the presence of (**B**) Cx32 CL2 peptide, (**C**) Cx35 CL2 peptide, (**D**) Cx43 CL2 peptide, (**E**) Cx45 CL2 peptide and (**F**) Cx57 CL2 peptide. The error bars represent SD.

**Figure 6 ijms-24-04153-f006:**
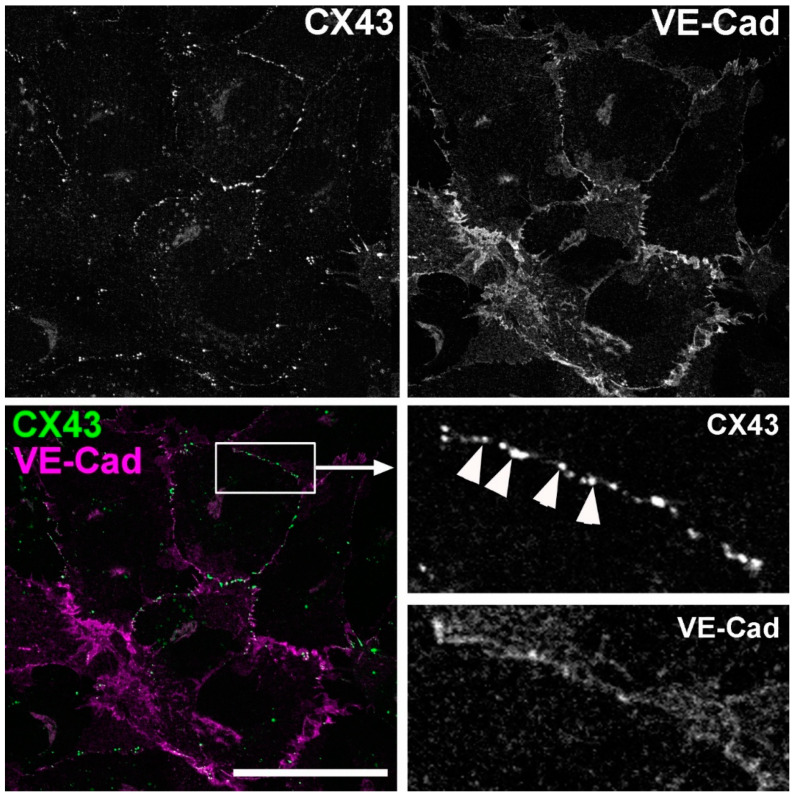
Cx43 positive localisation at cell–cell contact in cultured HUVEC. Confocal fluorescence images of HUVEC immune-labelled for endogenous Cx43 (1:25 dilution, top left and green in colour merged image) and VE-Cadherin (1:400 dilution, top right and magenta in colour merge image). Region indicated by white box in colour merge image is shown in grey scale on an expanded scale to right (pointed to by the arrow). Arrow heads indicate Cx43 puncta at cell–cell contacts. Images were acquired using a BioRad Radiance 2100 confocal microscope equipped with a 60 × 1.40 NA oil immersion objective and are representative of 6 independent experiments. Scale bar is 500 μm.

**Figure 7 ijms-24-04153-f007:**
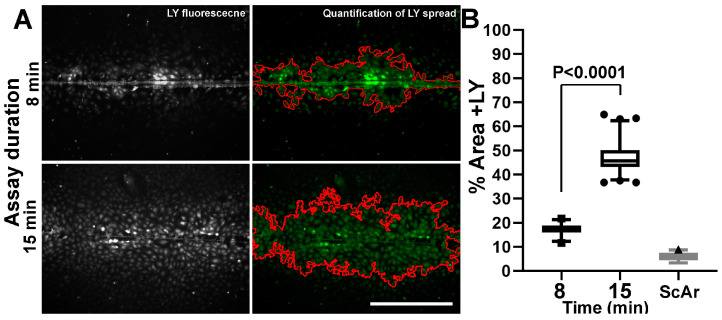
Confluent HUVEC monolayers are functionally gap junction coupled. (**A**) left panels: fluorescence images of HUVEC monolayers exposed to dye transfer buffer for 8 or 15 min as indicated, prior to fixation and imaged using an EVOS M5000 microscope equipped with a 10× 0.3 NA air objective. Right panels show the area of LY spread (red lines) in the images determined by the automated macro Cut Analyzer.ijm [48]. (**B**) Quantification of the time-dependent increase in area of LY spread fluorescence following 8 (*n* = 37 measurements from 3 independent experiments) or 15 min (*n* = 63 measurements from 3 independent experiments) after scratch formation. The mean area of the physical scratch lesion is shown for reference (ScAr). Data are expressed as box and whisker plots with median (horizontal bar), 25–75 percentile (box) and 5–95 percentiles (vertical bars). *p* < 0.0001, *t*-test, GraphPad Prism. Scale bar is 500 μm.

**Figure 8 ijms-24-04153-f008:**
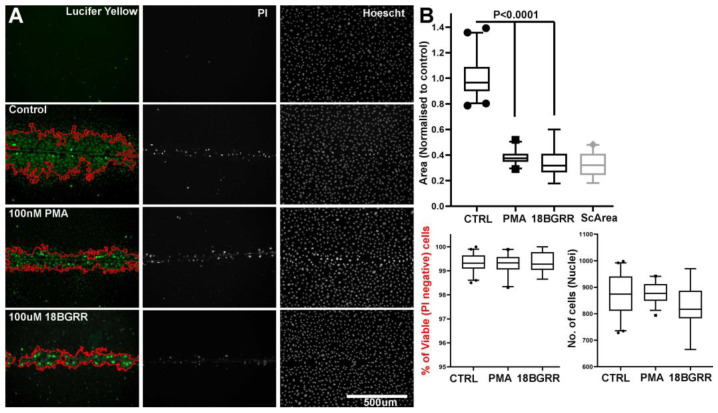
Inhibition of gap junctional coupling in HUVEC by PMA and 18 BGRR. (**A**) Left panels show LY signal in unscratched cells (top), vehicle control (DMSO) cells (second panel), PMA (100 nM; third panel) or 18BGRR (100 µM; bottom panel) treated cells 15 min post-scratch wound. Middle panels show PI staining and right panels nuclear staining by Hoechst. Red lines in LY images show the area of LY spread determined by the automated macro Cut_Analyzer.ijm [48]. (**B**) (top) the quantification of LY dye spread (normalised to the control) in control (*n* = 40 measurements), PMA (100 nM, *n* = 34) and 18GRR (100 µM, *n* = 17). Right (bottom) shows the cell count/field of view and cell viability (% of PI+ cells). The data are pooled from two independent experiments and are identical to the results obtained from a further two independent experiments (not shown). *p* < 0.0001 comparison to control, ANOVA, GraphPad Prism. Data are expressed as in Figure 7. Scale bar is 500 μm.

**Figure 9 ijms-24-04153-f009:**
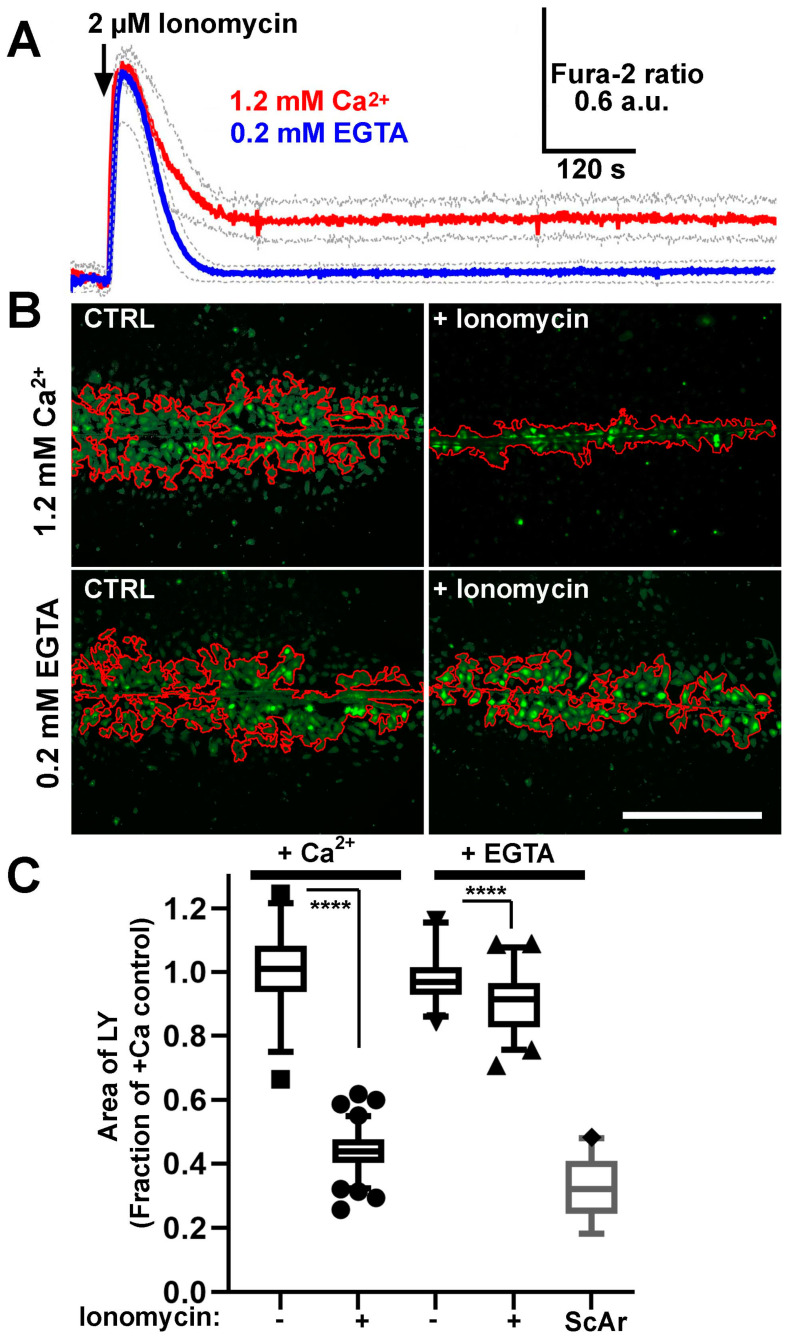
Ca^2+^-inhibition of gap junction coupling in HUVEC. (**A**) changes in 340/380 nm fura-2 fluorescence ratio (a.u. stands for arbitrary units) in HUVEC during stimulation with ionomycin (2 µM) in the presence or absence of external Ca^2+^, as indicated. Solid lines are the mean of 9 (plus Ca^2+^; red) and 11 (minus Ca^2+^; blue) cells, grey dashed lines show the 95% confidence limits. (**B**) representative images showing the spread of LY 8 min after lesion formation following an 8-min pre-exposure to vehicle control (left; 0.001% DMSO) or ionomycin (right) with or without external Ca^2+^. Red lines in images show the area of LY spread. (**C**) quantification of LY spread normalised to vehicle in control plus Ca^2+^ (*n* = 9), ionomycin plus Ca^2+^ (*n* = 8), control minus Ca^2+^ (*n* = 17), ionomycin minus Ca^2+^ (*n* = 14) treated cells respectively. The area of the scratch lesion (ScAr) is also indicated (*n* = 29). The data are pooled from two independent experiments. **** *p* < 0.0001 comparison to controls, ANOVA, GraphPad Prism. Data are expressed as for Figure 7. Scale bar is 500 μm.

**Figure 10 ijms-24-04153-f010:**
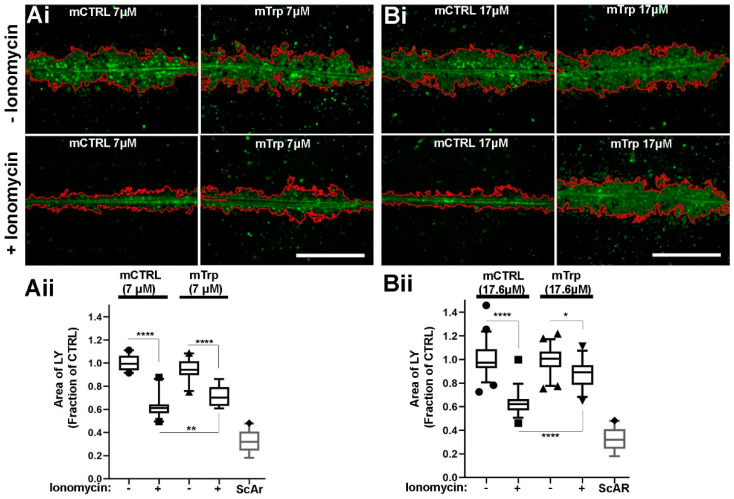
mTrp peptide reverses Ca^2+^-dependent GJ uncoupling in HUVEC. Ai-Bi, representative images showing LY spread 8 min after lesion formation following an 8-min pre-exposure to 7 μM (**Ai**); or 17 μM (**Bi**) mCtrl; or mTrp peptide with (bottom panels); or without (top panels) ionomycin, as indicated. Red lines in images show the area of LY spread. (**Aii**,**Bii**) quantification of LY spread normalised to control peptide minus ionomycin for data in (**Ai**,**Bi**). The data are pooled from two independent experiments. Each experiment was analysed by ANOVA in GraphPad Prism. **** *p* < 0.0001, ** *p* = 0.0073, * *p* = 0.041. Data are expressed as for Figure 7. The area of the scratch lesion (ScAr) is also indicated (*n* = 29). Scale bar 500 µm.

**Figure 11 ijms-24-04153-f011:**
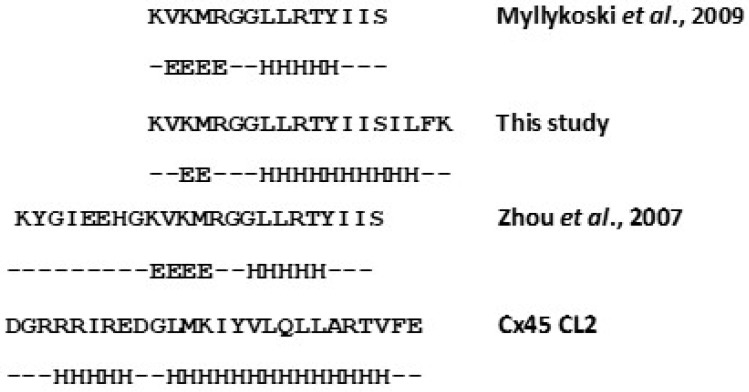
Cx43 CL2 peptide sequences used in various studies (Myllykoski et al., 2009 [52] and Zhou et al., 2007 [53]). Secondary structure prediction was performed using JPred [54]; E: extended; H: Helical). A similar break in the helical prediction exists for all our Cx CL2 peptides (see Cx45 CL2 peptide as an example).

**Table 2 ijms-24-04153-t002:** Interaction of Cx CL2 peptides with TA-CaM and DA-CaM in the presence of Ca^2+^.

Connexin	DA-CaM Quenching (%)	*K*_d_ (nM)	*^a^ K*_d_ for Ca^2+^ (nM)	Hill Coefficient *n*
	Ca^2+^	Ca^2+^/TA-CaM	TA-CaM	TA-CaM
TA-CaM	N.A.	N.A.	556 ± 6	5.3 ± 0.3
Cx32 CL2	65 ± 3	40 ± 4	311 ± 9	2.3 ± 0.1
Cx35 CL2	62 ± 6	31 ± 2	392 ± 7	2.6 ± 0.1
Cx43 CL2	61 ± 3	18 ± 1	340 ± 7	4.0 ± 0.2
Cx45 CL2	70 ± 5	75 ± 4	75 ± 17 597 ± 19	1 4.9 ± 0.6
Cx57 CL2	56 ± 4	60 ± 6	32 ± 4 1510 ± 96	1 0.96 ± 0.08
mTrp	82 ± 5	*^b^* 0.015	219 ± 5	1.3 ± 0.1

*^a^ K*_d_ for Ca^2+^ corresponds to the [Ca^2+^] at which TA-CaM reaches the half maximal fluorescence change during titration with Ca^2+^ in the presence of Cx CL2 peptide. *^b^* Calculated from kinetic measurements [24].

**Table 3 ijms-24-04153-t003:** Kinetic parameters of Ca^2+^/DA-CaM interaction with Cx CL2 peptides.

Connexin	Dissociation Kinetics *k*_off_ (s^−1^)		*K*_1_ (μM^−1^)	*k*_+2_ (s^−1^)	*k*_−2_ (s^−1^)
Cx32 CL2	14.5 ± 2.6 (62%)	1.8 ± 0.4 (38%)	1.45 ± 0.23	64.5 ± 3.7	14.5 ± 2.6
Cx35 CL2	7.7 ± 0.1 (54%)	2.6 ± 0.1 (46%)	2.07 ± 0.78	170 ± 15	7.7 ± 0.1
Cx43 CL2	0.35 ± 0.01 (46%)	0.13 ± 0.01 (53%)	3.2 ± 1.7	360 ± 38	0.35 ± 0.01
Cx45 CL2	15 ± 3 (14%)	4.0 ± 0.2 (86%)	*^a^* 216 ± 6	N.D.	13 ± 6
Cx57 CL2	57.5 ± 1.3 (79%)	6.1 ± 0.5 (21%)	0.55 ± 0.16	607 ± 86	57.5 ± 1.3

*K*_1_, *k*_+2_ and *k*_−2_ values represent fitted parameters to Equation in Figure 3R. *^a^* The slope in Figure 3K corresponds to *k*_+2_
*K*_1_ (μM^−1^s^−1^) of Equation in Figure 3R.

**Table 4 ijms-24-04153-t004:** The effect of TFE on the on-kinetics of Cx43 and Cx57 CL2 peptides.

Cx CL2	*k*_obs1_ *^a^* ↓ (s^−1^)	*k*_obs2_ *^b^* ↑ (s^−1^)
5 μM Cx43	0.0044 ± 0.0003	0
5 μM Cx43 + 15% TFE	1.82 ± 0.03	0.11 ± 0.01
17.5 μM Cx43	0.064 ± 0.002	0.0083 ± 0.0002
17.5 μM Cx43 + 15% TFE	2.36 ±0.02	0.020 ± 0.003
5 μM Cx57	N.A.	N.A.
5 μM Cx57 + 15% TFE	0.009 ± 0.001	N.A.
30 μM Cx57	0.0015 ± 0.0001	N.A.
30 μM Cx57 + 15% TFE	0.91 ± 0.03	N.A.

*^a^* Downward arrow indicates a phase in which DA-CaM fluorescence is decreased. *^b^* Upward arrow indicates a phase in which DA-CaM fluorescence is increased.

## Data Availability

Data supporting reported results are available on request.

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
