# Peer review of "Ca2+-Dependent and -Independent Calmodulin Binding to the Cytoplasmic Loop of Gap Junction Connexins"

_ijms, 2023, doi:10.3390/ijms24044153_

Round 1

Reviewer 1 Report

This is a carefully executed study to probe the biophysical parameters and Ca2+ dependence of the interactions between calmodulin (CaM) and multiple cytoplasmic loop-2 (CL2) connexin (Cx) peptides,  complemented by a functional assay in living cells. Fluorescence spectroscopy and fast stopped-flow fluorimetry at physiological conditions are used to study interactions (affinity and binding kinetics) between CaM and CL2 peptides from selected connexins of the alpha, beta, gamma family. Of particular interest are results suggesting that for two Cxs, Ca2+-dependent CaM association may occur at resting Ca2+i (50-100 nm) supporting very high Ca2+ affinity of one CaM Ca2+-bdg site in the complex. The effect of the Ca2+-CaM-Cx interaction on gap junction permeability is determined in HUVEC cells and corroborated by using a cell-permeable high-affinity “CaM sponge” (mTrp peptide). 

There are numerous relatively minor issues that require the authors’ attention, as detailed below.

1. Line 72: …at 1:400 dilution (DAKO) at 1:250 dilution: Unclear sentence; please correct.

2. Line 109: microscope where and maintained…; Lines 123, 142: …central part of the of each well…; Line 127: …experiment and was the plates were…: Please correct. Line 145: processed.

3. Lines 161-163 …and are described below: The supplemental data should be mentioned here.

4. Line 212: Ref. 21 is not the proper citation for TA-Cl conjugation to Lys75 or the TA-CaM scheme.

5.Table 2 (line 217): Indicate n=Hill coefficient.

6. Lines 254, 262: Table 3 - not Table 2.

7. Line 282 (Figure 3 legend): …equation (R) derived from the model shown in scheme (P).

8. Line 316: This should be Table 4, and the Table should be mentioned in the text of the paragraph on lines 307-315.

9. Line 408/Figure 7: Measurements were apparently taken 8 and 15 min after scratching/LY application. This should be more clearly mentioned in the Methods (lines 117-134).

10. Line 424: …coupled we (not We) next…

11. Figure 8A, label in top right panel: Hoechst (not Hoescht).

12. Figure 10A: indicate the peptide concentrations (7µM and 17.6 µM) on top of the panels or in the legend (lines 495ff). Also, the peptide sequences on top of panels A should end in CONH2 (not COO2) as they were C-terminal amides not free acids.

13. Figures 6,7, 9,10: Length of the scale bar should be indicated in the legend (it is only shown in Fig. 8).

14. Figures 2-10 appear to be shifted to the left; the panel lettering is also shifted.

15. The supplementary figure 1 should be mentioned at the proper place in the text.

16. Line 518: Figure 11 should be mentioned in the text. In the figure, the reference numbers should be indicated for Myllykoski et al. 2009 [54] and Zhou et al. 2007 [55].

17. Line 572: …75 and 32 nM,…

18. References: Ref 5: volume, page numbers? Ref 22: incomplete citation,  Int J Mol.Sci 21(14), 4938.

Ref 34: incomplete citation, Front Mol Neurosci 9, 120. Ref 47: incomplete citation, Sci Rep 10(1), 730.

Ref 51: vol 38, pg 3936, 1999 at the end of the title? Please correct. Ref 57: …binds to (not finds to).

A general comment: Based on kinetic and binding affinity results of the CaM-connexin interactions, the authors propose an interesting hypothesis for Ca2+-regulated, binding-induced structural changes occurring in gap junction connexins to regulate pore opening/closing. Obviously, the proposed structural change model is  speculative given that the biophysical/biochemical data were obtained with relatively short Cx peptides rather than with the intact molecular complex in situ. However, functional studies using the dye spread assay and a high-affinity CaM “sponge” peptide support the physiological relevance of the studied interactions. Further structural studies on the complexes in the presence and absence of Ca2+ may well shed further light on the complex issue of gap junction regulation by Ca-CaM.

Author Response

Please see our response in attached file

Reviewer 2 Report

1. The researchers investigate the interactions between several connexin isoforms with the Calmodulin system. They analyzed several of these binding interactions and extracted important conclusions to explain opening of gap junctions on response to physiological stimulus. 2. The topic is not original on the field, since several other papers address interactions between connexins and other cellular proteins but it is really relevant. Connexins are truly complex proteins that oligomerize to form hexamers (hemichannels) and 2 hemichannels from different cells dock together to form a gap junction channel. Interactions with these proteins regulate cell homoeostasis under different stimulus. 3. This particular work adds knowledge about connexins isoforms from 3 different families and their interactions with Calcium and the Calcium Calmodulin system. Calcium binding per se regulates the opening/closure of connexins hemichannels and gap junctions. 4. On my opinion the methodology used by the authors is correct to prove the points addressed on the paper. 5. The conclusions are consistent with the results and they address perfectly the question posted. 6. References used are appropriate. 7. Tables look good, my only comment is to be careful at final version of figures with the reference letters for the panels in each figure.

Author Response

(The authors gave the same response as above.)

Reviewer 3 Report

In this manuscript, the authors report the mechanism of interaction between Ca2+/CaM and the cytosolic loops of gap junction Cx. They investigated the binding of CaM with Cx with or without the presence of Ca ions. The authors also investigated the Ca/CaM-dependent effect on gap junction in HUVEC. 

I have a few questions and suggestions based on the results in the manuscript:

Major:

1. The authors used only the fluorescence-based method for all the binding studies between Cx and CaM. Since it is a novel method used in this paper, i suggest the authors use another binding assay, such as iTC or SPR, to validate the reliability of the method. iTC or SPR results of one or two Cx-CaM will be enough to validate the reliability of the method. 

2. In figure 3/4, the authors show the curves of the fluorescence quenching, which is only one curve for each concentration. Please do and show at least 3 repeats for each concentration. 

3. For the cell assay part of the paper, what is the novelty of the results? It is known that the Ca/CaM can regulate the gap junction through interaction with Cx. 

4. The manuscript presents all the binding results between CaM with the loop on Cx, which is only a small part of the Cx protein. The structures of multiple Cx were solved in the last a few years. I wonder if the interaction can still be detected with full-length Cx protein. 

Author Response

(The authors gave the same response as above.)

Round 2

Reviewer 3 Report

All questions are answered. I have no more comments on the manuscript.